# Augmenting Molecular Graphs with Geometries via Machine Learning Interatomic Potentials

**Cong Fu**[*]                                               *congfu@tamu.edu*
*Department of Computer Science & Engineering*
*Texas A&M University*

**Yuchao Lin**[*]                                            *kruskallin@tamu.edu*
*Department of Computer Science & Engineering*
*Texas A&M University*

**Zachary Krueger**                                  *zacharykrueger321@tamu.edu*
*Department of Computer Science & Engineering*
*Texas A&M University*

**Haiyang Yu**                                               *haiyang@tamu.edu*
*Department of Computer Science & Engineering*
*Texas A&M University*

**Maho Nakata**                                             *maho@riken.jp*
*RIKEN Cluster for Pioneering Research, RIKEN*

**Jianwen Xie**                                            *jianwen.xie@lambdal.com*
*Lambda, Inc.*

**Emine Kucukbenli**                                      *ekucukbenli@nvidia.com*
*NVIDIA*

**Xiaofeng Qian**                                            *feng@tamu.edu*
*Department of Materials Science & Engineering*
*Texas A&M University*

**Shuiwang Ji**                                              *sji@tamu.edu*
*Department of Computer Science & Engineering*
*Texas A&M University*

**Reviewed on OpenReview:** *https://openreview.net/forum?id=JwxhHTISJL*

## Abstract

Accurate molecular property predictions require 3D geometries, which are typically obtained using expensive methods such as density functional theory (DFT). Here, we attempt to obtain molecular geometries by relying solely on machine learning interatomic potential (MLIP) models. To this end, we first curate a large-scale molecular relaxation dataset comprising 3.5 million molecules and 300 million snapshots. Then MLIP pre-trained models are trained with supervised learning to predict energy and forces given 3D molecular structures. Once trained, we show that the pre-trained models can be used in different ways to obtain geometries either explicitly or implicitly. First, it can be used to obtain approximate low-energy 3D geometries via geometry optimization. While these geometries do not consistently reach DFT-level chemical accuracy or convergence, they can still improve downstream performance compared to non-relaxed structures. To mitigate potential biases

---

[*]Equal contribution

and enhance downstream predictions, we introduce geometry fine-tuning based on the relaxed 3D geometries. Second, the pre-trained models can be directly fine-tuned for property prediction when ground truth 3D geometries are available. Our results demonstrate that MLIP pre-trained models trained on relaxation data can learn transferable molecular representations to improve downstream molecular property prediction and can provide practically valuable but approximate molecular geometries that benefit property predictions. Our code is publicly available at: `https://github.com/divelab/AIRS/`.

## 1 Introduction

Molecular property prediction is a critical task in drug discovery, chemistry, and materials science (Zhang et al., 2023; Liyaqat et al., 2024). Many molecular properties are strongly influenced by the stable 3D structure of a molecule, corresponding to its lowest potential energy configuration. For example, as shown in Table 1, in predicting the HOMO-LUMO gap property, GIN (Hu et al., 2021)—which uses only 2D molecular graphs as input—achieves notably worse performance compared to PaiNN (Schütt et al., 2021), which leverages stable 3D geometries. With accurate 3D structures, 3D geometric neural networks (3DGNNs) can significantly improve property prediction accuracy. However, the current standard for obtaining stable molecular structures relies on computationally expensive methods such as density functional theory (DFT) for geometry optimization. Uni-Mol+ (Lu et al., 2023) attempts to bridge this gap by predicting stable 3D geometries during training, allowing only non-stable molecule structures to be used during testing; however, it outperforms GIN but still exhibits a significant performance gap compared to 3DGNNs, highlighting that obtaining useful 3D geometries for property prediction remains a major challenge.

To bridge the gap of obtaining 3D geometries for molecular property prediction, we aim to train a machine learning interatomic potential (MLIP) pre-trained model to assist geometry relaxation for downstream tasks where only non-stable molecular structures are available during testing. This approach enables the use of downstream 3DGNNs for property prediction using pre-trained model-relaxed geometries, referred to as Force2Geo in Table 1. We emphasize that Force2Geo produces approximate geometries that may not always converge to the true DFT-optimized structures, and its effectiveness depends on the molecular system and down-

Table 1: Performance gap between 2D and 3D models for HOMO-LUMO gap prediction on Molecule3D dataset.

| Model | Validation MAE (eV) |
|---|---|
| GIN | 0.1249 |
| Uni-Mol+ | 0.1070 |
| Force2Geo + PaiNN | 0.0794 |
| DFT + PaiNN | 0.0562 |

stream task. Additionally, the pre-trained model can be directly fine-tuned on downstream tasks when 3D molecular structures are provided. pre-trained models have demonstrated remarkable success in computer vision (Bommasani et al., 2021; Liu et al., 2024) and natural language processing (Brown et al., 2020; Touvron et al., 2023), where pre-training yields transferable representations that significantly enhance downstream performance. However, the development of MLIP pre-trained models for small molecules has been hindered by the lack of large-scale datasets with DFT-level accurate energy and force labels.

In this work, to address the challenge of efficiently obtaining accurate 3D molecular structures, we curate a large-scale molecular relaxation dataset comprising 3.5 million small molecules and 300 million snapshots with energy and force labels, including 105 million snapshots computed using DFT at the B3LYP/6-31G* level of theory. By leveraging this extensive dataset, we can train an MLIP pre-trained model that can be used in geometry optimization to obtain computationally efficient but approximate 3D geometries, referred to as Force2Geo, providing a cost-effective alternative to conventional quantum methods such as DFT. The choice of backbone models for pre-training can be found in Section 4.1. Additionally, we introduce geometry fine-tuning to enhance the downstream predictive accuracy of 3DGNNs using relaxed 3D structures. Furthermore, the MLIP pre-trained model can be directly fine-tuned for property prediction when ground truth 3D geometries are available, termed Force2Prop, extending its applicability to a range of downstream tasks.

Our contributions are threefold:

- We curate a large-scale molecular relaxation dataset with 3.5M molecules and 300M snapshots, including 105M DFT-level energy and force labels, enabling MLIP model pre-training.

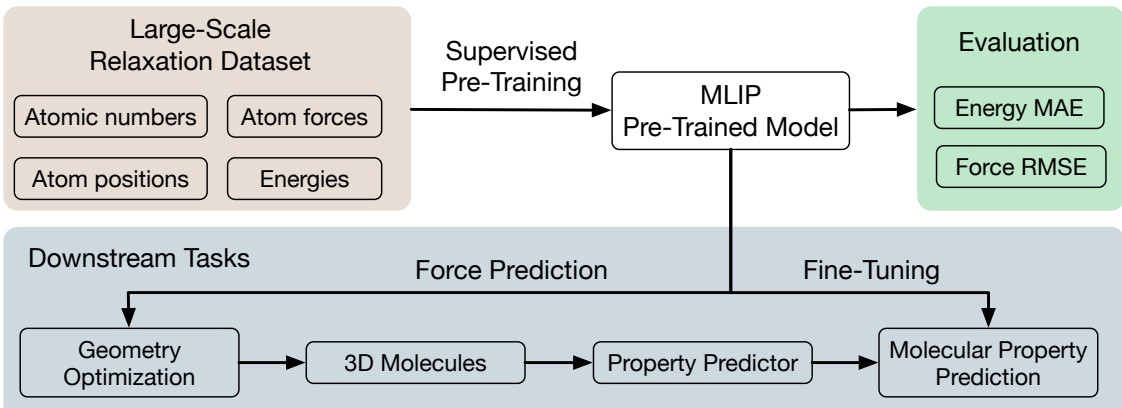

Figure 1: Overview of the MLIP pre-trained model training pipeline. The model is pre-trained using our curated large-scale relaxation dataset, which includes atomic numbers, forces, positions, and energies for each snapshot. The pre-trained MLIP model can either be fine-tuned for molecular property prediction when stable 3D geometries are available or employed for geometry optimization to obtain 3D geometries for downstream property prediction.

- We show that the pre-trained MLIP model on our dataset can efficiently produce low-energy 3D geometries via geometry optimization for downstream property prediction, and introduce geometry fine-tuning to further improve 3D GNN performance.

- We demonstrate that the pre-trained model can be directly fine-tuned for molecular property prediction when ground-truth 3D geometries are available, highlighting the effectiveness of pre-trained MLIPs in supporting diverse downstream tasks.

## 2 Method

In Section 2.1, we present our curated large-scale relaxation dataset. In Section 2.2, we describe the MLIP pre-trained model. In Section 2.3, we outline the geometry optimization process using the pre-trained model. Finally, in Section 2.4, we discuss geometry fine-tuning for molecular property prediction on MLIP pre-trained model-relaxed structures.

### 2.1 Large-Scale DFT Relaxation Dataset

To train an MLIP pre-trained model, a large-scale relaxation dataset with energy and force labels is essential. However, such a dataset for small molecules is currently unavailable. To address this gap, we curated PubChemQCR (Fu et al., 2025), a new dataset containing DFT-based geometry optimization trajectories for approximately 3.5 million molecules. These molecules are sourced from the PubChem Compound database. The raw trajectory data originate from the PubChemQC database (Nakata & Shimazaki, 2017), and molecular relaxation is performed sequentially using PM3 semi-emperical method, Hartree-Fock, and DFT at the B3LYP/6-31G* level. We extracted atomic numbers, energies, and atomic forces for each snapshot, resulting in 3,471,000 trajectories and 298,751,667 molecular snapshots, of which 105,494,671 snapshots are DFT-calculated. On average, each molecule contains 29 atoms, including 14 heavy atoms. Further dataset details can be found in Appendix A and PubChemQCR (Fu et al., 2025).

### 2.2 MLIP pre-trained Models for Small Molecules

Machine learning interatomic potentials (MLIPs) are designed to approximate the potential energy surface (PES) of molecular systems, which is traditionally computed using quantum mechanical methods such as density functional theory (DFT). These methods are computationally intensive, motivating the use of MLIPs

to learn the PES from DFT-calculated data. The total energy $E$ is predicted based on atomic coordinates $\{\boldsymbol{x}_i\}_{i=1}^N$ and atomic numbers $\{\boldsymbol{a}_i\}_{i=1}^N$, often decomposed as a sum over atom-wise contributions, $E = \sum_i E_i$. To ensure energy conservation, atomic forces are obtained as the negative gradient of the predicted energy with respect to atomic positions, $\boldsymbol{f}_i = -\nabla_{\boldsymbol{x}_i} E$. The pipeline of the training and usage of the MLIP pre-trained model is shown in Fig. 1. The MLIP pre-trained model enables efficient geometry optimization to obtain 3D geometries for downstream predictors requiring 3D molecular structures as inputs. Additionally, by capturing the underlying physics of atomic interactions during pre-training, the MLIP pre-trained model learns informative molecular representations that can be directly fine-tuned for various downstream tasks.

To effectively serve as an MLIP pre-trained model, a suitable backbone architecture is required to encode molecular information and learn geometric relationships. For molecule representation, we usually represent molecules as graphs $\mathcal{G} = \{V, X, A\}$, where $V \in \mathbb{R}^{n \times d}$ denotes node features, $X \in \mathbb{R}^{n \times 3}$ represents the 3D coordinates of atoms, and $A \in \{0, 1\}^{n \times n}$ is the adjacency matrix. In 3D molecular graphs, edges are often constructed using a radius graph, where an edge is formed between two atoms if their Euclidean distance is within a predefined cutoff. Molecules that share the same chemical graph but differ in their 3D coordinates $X$ are referred to as *conformers*. For backbone models, geometric neural networks are well-suited as they are designed for learning on data with underlying spatial or geometric structures, and are widely used for modeling molecular systems. In this work, we consider existing geometric neural networks as candidate backbones without developing new architectures, as architecture design is outside the scope of this study.

Formally, given a 3D molecular graph $\mathcal{G} = \{V, X, A\}$, where each node has a feature vector $\boldsymbol{v}_i \in \mathbb{R}^d$ and position $\boldsymbol{x}_i \in \mathbb{R}^3$, and each edge has a feature $\boldsymbol{a}_{ij} \in \mathbb{R}^d$, a general message passing layer for molecular systems at layer $l$ is defined as:

$$\boldsymbol{m}_{ij}^{(l)} = \phi_m^l \left( \boldsymbol{v}_i^{(l)}, \boldsymbol{v}_j^{(l)}, \boldsymbol{x}_i, \boldsymbol{x}_j, \boldsymbol{a}_{ij} \right), \tag{1}$$

$$\boldsymbol{v}_i^{(l+1)} = \phi_u^l \left( \boldsymbol{v}_i^{(l)}, \mathrm{AGG} \left( \left\{ \boldsymbol{m}_{ij}^{(l)} \mid j \in \mathcal{N}(i) \right\} \right) \right), \tag{2}$$

$$\boldsymbol{y} = \mathrm{READOUT} \left( \left\{ \boldsymbol{v}_i^{(L)} \mid i \in V \right\} \right), \tag{3}$$

where $\phi_m$ and $\phi_u$ are neural networks, and $\mathrm{AGG}(\cdot)$ denotes a permutation-invariant aggregation function such as `mean`, `sum`, or attention-based mechanisms. After the final message passing layer, a readout function is applied to aggregate the node embeddings into a graph-level representation $\boldsymbol{y}$ for molecular property prediction. Alternatively, node embeddings can be used for node-level prediction such as predicting atom-wise energy. In molecular applications, geometric neural networks preserve equivariant or invariant representations to respect symmetries like rotation and translation, requiring $\phi_m$ and $\phi_u$ to be equivariant functions and $\boldsymbol{v}_i^{(l)}$ to be geometric objects (*e.g.* vectors or tensors).

Since the dataset includes trajectories computed at varying levels of quantum accuracy, we use only the snapshots from the DFT substage to train the MLIP pre-trained model, leaving the exploration of training with mixed levels of accuracy for future work. Additionally, the DFT substage is the most computationally intensive, taking several hours per molecule, whereas the PM3 and Hartree-Fock stages require only a few minutes. The training objective for the MLIP pre-trained model includes both energy and force prediction, formulated as:

$$\mathcal{L} = \lambda_E \cdot \mathcal{L}_E + \lambda_F \cdot \mathcal{L}_F, \tag{4}$$

$$\mathcal{L}_E = \frac{1}{N} \sum_{i=1}^N |\hat{e}_i - e_i|, \tag{5}$$

$$\mathcal{L}_F = \sqrt{\frac{1}{M} \sum_{j=1}^M \|\boldsymbol{f}_j - \hat{\boldsymbol{f}}_j\|^2}, \tag{6}$$

where N denotes the number of molecules in a batch, M denotes the number of atoms in a batch, $\lambda_E$ and $\lambda_F$ are weights to balance the energy and force loss terms, $\hat{e}_i$ and $e_i$ are the predicted and ground truth energies, and $\hat{\boldsymbol{f}}_j$ and $\boldsymbol{f}_j$ are the predicted and ground truth forces for each atom.

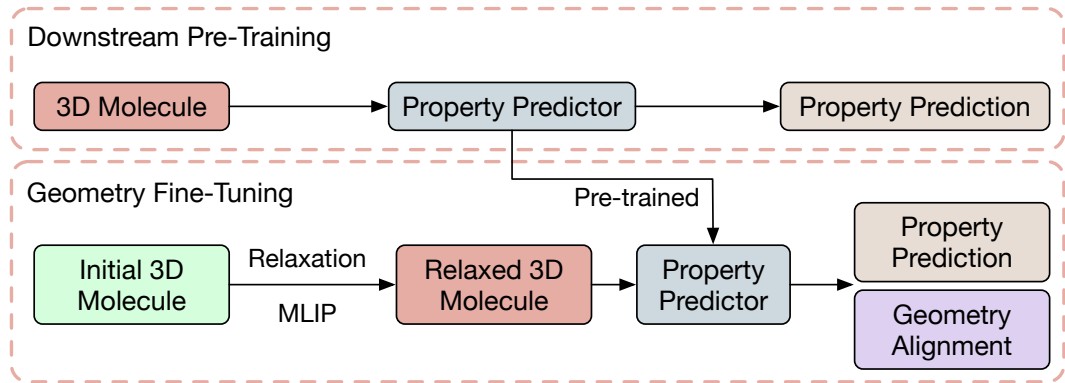

Figure 3: Overview of geometry fine-tuning. In the pre-training stage, a property predictor is trained on stable 3D molecules for property prediction. This pre-trained predictor is then fine-tuned on 3D molecular structures relaxed by the MLIP pre-trained model, with both property prediction and geometry alignment losses.

## 2.3 Geometry Optimization

After training the MLIP pre-trained model, it can be employed to perform geometry optimization to obtain stable 3D molecular structures. The objective of geometry optimization is to adjust the atomic positions to minimize the potential energy of the system while adhering to predefined convergence criteria.

As shown in Fig. 2, the conventional DFT-based relaxation (Nakata & Shimazaki, 2017) process begins by selecting an exchange-correlation energy functional and a basis set. The electronic structure is then determined iteratively using the self-consistent field (SCF) method. Once the SCF loop converges, the energy of the molecule and the corresponding atomic gradients are calculated. These gradients are then used to update atomic positions through optimization algorithms, such as Newton's method. This process is repeated until the maximum force falls below the predefined convergence threshold. In the MLIP-based relaxation process, the neural network predicts the forces directly, replacing the DFT calculation, while the remaining procedure, including the optimization loop, remains unchanged.

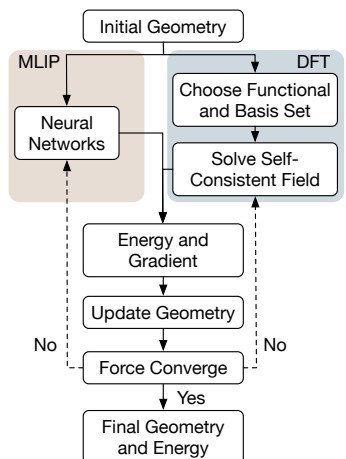

Figure 2: Comparison of geometry optimization based on DFT and MLIP.

In this work, we utilize the Broyden–Fletcher–Goldfarb–Shanno (BFGS) (Fletcher, 2000) as the optimization algorithm for geometry optimization. BFGS is a quasi-Newton method that approximates the Hessian matrix to efficiently update atomic coordinates and reduce the overall computational cost. The optimization process iteratively adjusts the atomic coordinates $\{\boldsymbol{x}_i\}_{i=1}^N$ to minimize the potential energy $E(\boldsymbol{x})$ predicted by the MLIP pre-trained model. At each step, the atomic forces $\boldsymbol{f}_i = -\nabla_{\boldsymbol{x}_i} E$ are computed to guide the atomic displacements. The maximum force is calculated as:

$$\mathrm{max\_force} = \max\left(\|\boldsymbol{f}_i\|\right), \tag{7}$$

where $\boldsymbol{f}_i \in \mathbb{R}^3$ is the force vector for atom $i$ and $\|\cdot\|$ denotes L2 norm. The optimization process terminates when one of the following stopping criteria is met:

- The maximum force, as defined above, falls below a threshold of $0.05\,\mathrm{eV/\mathring{A}}$.

- The maximum number of optimization steps is reached, set to 500.

### 2.4 Geometry Fine-Tuning

As discussed in Section 1, when accurate 3D molecular structures are available in the test set, 3DGNNs can predict molecular properties more effectively than Uni-Mol+. The MLIP pre-trained model can be used to obtain 3D geometries through geometry optimization. However, using pre-trained model-relaxed conformers may introduce errors and biases due to imperfect force prediction. To address this, we introduce geometry fine-tuning to improve the accuracy of 3DGNN predictions based on these relaxed structures. The whole pipeline of geometry fine-tuning is shown in Fig. 3. Specifically, we first pre-train a downstream predictor on the downstream dataset using ground-truth 3D geometries as input. We then relax the downstream training molecules—starting from unstable conformers—using the pre-trained MLIP model, and fine-tune the downstream predictor on these relaxed structures. This allows the predictor to adapt to the geometric distribution produced by the pre-trained model. During testing, only the pre-trained model-relaxed geometries are provided to the downstream predictor.

For geometry fine-tuning, we adopt a multi-task learning framework by introducing geometry alignment as an auxiliary task to support the primary objective of property prediction. This auxiliary task is motivated by two key intuitions: (1) learning to predict deviations from the ground-truth geometry encourages the model to attend to subtle 3D structural cues that are critical for accurate property estimation; and (2) during fine-tuning, the model is exposed to relaxed geometries generated by the pre-trained model, which inevitably differ from the ground-truth geometries seen during pre-training. This introduces a distribution shift, and the auxiliary geometry alignment task helps bridge this gap by encouraging the model to relate the relaxed geometries back to the original ground-truth domain.

To implement the auxiliary task, we adopt a mixed conformer denoising strategy. During training, half of the input structures are ground-truth conformers with added coordinate noise, and the other half are randomly sampled from the relaxation trajectory produced by the MLIP pre-trained model. In addition to the main property prediction loss, we include a geometry alignment loss based on cosine similarity, which encourages alignment between the predicted and target atomic displacements. Formally,

$$\mathcal{L}_{\text{total}} = \mathcal{L}_{\text{prop}} + \lambda \cdot \mathcal{L}_{\text{geo}}, \tag{8}$$

where $\mathcal{L}_{\text{prop}}$ is the loss for molecular property prediction and $\mathcal{L}_{\text{geo}}$ is the auxiliary geometry alignment loss, defined as:

$$\mathcal{L}_{\text{geo}} = \frac{1}{N} \sum_{i=1}^{N} \left(1 - \cos\left(\Delta \hat{\mathbf{r}}_i, \Delta \mathbf{r}_i\right)\right), \tag{9}$$

where $\Delta \hat{\mathbf{r}}_i$ is the predicted displacement vector for atom $i$, $\Delta \mathbf{r}_i$ is the target displacement vector, and $\cos(\cdot, \cdot)$ denotes cosine similarity. $\lambda$ is a hyperparameter to weight the auxiliary loss.

## 3 Related Work

**Molecular Representation Learning.** Learning informative molecular representations is critical for accurately predicting molecular properties. Early efforts focused on pre-training models on SMILES strings using language modeling techniques such as BERT (Devlin et al., 2019), exemplified by SMILES-BERT (Wang et al., 2019). Subsequently, attention shifted toward pre-training on 2D molecular graphs by designing self-supervised learning tasks (Hu et al., 2019; Rong et al., 2020). Other works further explored learning shared representations between 2D and 3D molecular graphs through contrastive learning (Liu et al., 2021; Stärk et al., 2022). More recently, denoising pre-training (Zaidi et al., 2022; Feng et al., 2023; Ni et al., 2023; Liao et al., 2024) has emerged as a highly effective approach for molecular representation learning. In this paradigm, 3D molecular structures are perturbed with specific noise, and the model is trained to predict the applied noise. Denoising pre-training has been shown to outperform previous pre-training methods, as it is mathematically equivalent to learning an underlying interatomic potential, leading to richer and more physically grounded representations. Another line of work focuses on 3D GNNs that learn informative representations of molecular geometries to predict molecular properties. Representative methods include SchNet (Schütt et al., 2017), SphereNet (Liu et al., 2022), ComENet (Wang et al., 2022), DimeNet++

(Gasteiger et al., 2020), TorchMD-Net (Thölke & De Fabritiis, 2022), and PaiNN (Schütt et al., 2021), etc. A key limitation of these methods is that they require access to ground-truth 3D geometries in order to achieve accurate predictions.

**Molecular Foundation Models.** Inspired by the remarkable success of pre-trained models in computer vision (Bommasani et al., 2021; Liu et al., 2024; Zhai et al., 2022; Dehghani et al., 2023) and natural language processing (Brown et al., 2020; Touvron et al., 2023; Achiam et al., 2023; Bai et al., 2023), molecular pre-trained models have also begun to attract significant attention. However, due to the lack of large-scale 3D molecule data with energy and force labels, existing molecule pre-trained models primarily focus on 1D SMILES strings or 2D molecular graph representations. For example, ChemFM (Cai et al., 2024) adopts self-supervised causal language modeling on SMILES, while MolE (Méndez-Lucio et al., 2022) adapts DeBERTa (He et al., 2020) for molecular graphs. MolFM (Luo et al., 2023) jointly learns molecular representations from graphs, biomedical texts, and knowledge graphs. Nevertheless, none of these models address the fundamental task of learning 3D molecular energies and forces, which limits their applicability to geometry-dependent tasks such as 3D molecular property prediction, geometry optimization, etc.

## 4    Experiments

In this section, we demonstrate the effectiveness of the MLIP pre-trained model across several downstream tasks. In Section 4.1, we benchmark several backbone model candidates on PubChemQCR-S. In Section 4.2, we present geometry optimization using the pre-trained model. In Section 4.3, we show that pre-trained model-relaxed geometries can improve molecular property prediction when stable 3D structures are unavailable in the test set. In Section 4.4, we fine-tune the pre-trained model for 3D molecular property prediction.

### 4.1    Backbone Models

To select the backbone architecture for the MLIP pre-trained model, we benchmark several representative MLIP models on our curated dataset. The methods include SchNet (Schütt et al., 2018), FAENet (Duval et al., 2023), NequIP (Batzner et al., 2022), Equiformer (Liao & Smidt, 2022), SevenNet (Park et al., 2024), Allegro (Musaelian et al., 2023), PaiNN (Schütt et al., 2021), PACE (Xu et al., 2024), and MACE (Batatia et al., 2022).

Table 2: Performance of representative machine learning interatomic potential (MLIP) models on the PubChemQCR-S validation set.

| Model | Energy MAE (meV/atom) | Force RMSE (meV/Å) | Time (min/epoch) |
|---|---|---|---|
| SchNet | 5.30 | 56.55 | 25 |
| PaiNN | 5.13 | 46.34 | 26 |
| NequIP | 7.37 | 54.78 | 130 |
| SevenNet | 8.77 | 47.63 | 150 |
| Allegro | 10.86 | 60.71 | 85 |
| FAENet | 7.28 | 60.24 | 16 |
| MACE | 7.54 | 51.46 | 120 |
| PACE | 6.24 | 50.54 | 140 |
| Equiformer | **4.69** | **34.67** | 65 |

For training efficiency, we curated a smaller subset for model benchmarking, named PubChemQCR-S, which contains 40,979 trajectories and 1,504,431 molecular snapshots from the DFT stage calculations. The benchmark results are summarized in Table 2. In selecting a pre-trained model backbone, we consider both predictive performance and computational efficiency. Among the candidates, PaiNN demonstrates relatively strong energy prediction accuracy and the second-best force prediction performance on the benchmark while maintaining computational efficiency, making it a strong choice for large-scale pre-training. Detailed introduction and training of those benchmarked methods can be found in Appendices B and C.

### 4.2    Geometry Optimization

**Dataset.** To evaluate geometry optimization performance, we select 1,000 molecules from the test set of PubChemQCR-S using MaxMin diversity sampling to ensure maximal structural diversity. Specifically, we compute the Morgan fingerprint for each molecule in the test set and randomly select an initial molecule. We then iteratively add the molecule that has the largest Tanimoto distance from the currently selected set. We refer to the resulting geometry optimization test set as $\mathcal{D}_{\text{opt}}$.

**Metrics.** Since the pre-trained model is trained using data from the DFT relaxation stage, we evaluate its geometry optimization performance by relaxing molecules starting from the first snapshot of the DFT

trajectory. The evaluation metrics we adopt are partially based on those proposed in (Tsypin et al., 2023), and include: (1) **Average Energy Minimization Percentage**, $\overline{\text{pct}}_T$, quantifies how much energy is minimized by the MLIP-based optimization relative to the DFT-based optimization; (2) **Chemical Accuracy Success Rate**, $\text{pct}_{\text{success}}$, measures the percentage of relaxed molecules whose residual energy is within chemical accuracy (commonly defined as 1 kcal/mol); (3) **Divergence Rate**, $\text{pct}_{\text{div}}$, represents the percentage of relaxed molecules for which either the single-point DFT energy calculation failed or the relaxed DFT energy is higher than the initial energy; (4) **Force Convergence Rate**, $\text{pct}_{\text{FwT}}$, measures the percentage of relaxed molecules whose maximum force is below a threshold of 0.05 eV/Å. More details can be found in Appendix D.

**Results.** The results are presented in Table 3. While the value of $\overline{\text{pct}}_T$ indicates that the MLIP-relaxed geometries reduce a certain amount of energy compared to the initial structures, a notable portion of conformers remains outside the optimal region. Consequently, the values of $\text{pct}_{\text{success}}$ and $\text{pct}_{\text{FwT}}$ are relatively low. This highlights the inherent challenge of optimizing molecular geometries that are already near the energy minimum—a regime where achieving further relaxation requires extremely accurate modeling of the potential energy surface.

Table 3: Geometry optimization performance of the MLIP pre-trained model pre-trained on the curated PubChemQCR dataset.

| Model | $\overline{\text{pct}}_T$(%) | $\text{pct}_{\text{success}}$ (%) | $\text{pct}_{\text{div}}$ (%) | $\text{pct}_{\text{FwT}}$ (%) |
|---|---|---|---|---|
| Force2Geo | 57.37 | 10.29 | 8.1 | 4.2 |

**Discussions.** We observe that the performance of geometry optimization is not yet ideal, indicating substantial room for improvement. The challenge of achieving high-quality geometry optimization using near-optimal training data can be attributed to several factors. First, the training data predominantly resides in low-force regimes, which provide weak learning signals. In these regions, even though the model has seen similar configurations, the small forces make it difficult for the model to learn very accurate gradients. Second, MLIP-based relaxation becomes highly sensitive to small deviations in force direction when true gradients are small. Near energy minima, the model is required to predict very small force vectors with high precision—an inherently difficult task. Third, in low-gradient regions, the predicted forces often contain noise that can be significant relative to the magnitude of the true forces, further increasing the difficulty of accurate force prediction.

## 4.3 Molecular Property Prediction with MLIP Pre-Trained Model Relaxed Geometries

**Task.** As discussed in Section 1, obtaining accurate 3D geometries is crucial for achieving strong performance in quantum property prediction. However, acquiring ground-state geometries typically requires time-consuming DFT-based relaxation. To address this limitation, it is highly desirable to leverage machine learning interatomic potentials (MLIPs) to generate approximate stable geometries that can support downstream property prediction. In this task, we focus on a practical setting where ground-truth 3D geometries are available during training, but only non-stable geometries are available at test time—consistent with the setup used in Uni-Mol+.

**Dataset.** We use the Molecule3D dataset and predict the HOMO-LUMO gap, a key quantum property of molecular electronic structure. We use a subset of the dataset containing 600,000 molecules for *random* and *scaffold* splits. More details about the dataset can be found in Appendix E.

**Results.** We compare geometry fine-tuning using input structures with varying levels of geometric accuracy. Specifically, we evaluate downstream performance using: (1) RDKit-generated geometries, (2) geometries after the PM3 and HF optimization, i.e. geometries before the DFT relaxation, and (3) MLIP pre-trained model-relaxed geometries. We include (2) because the pre-trained model is trained on DFT substage trajectories and the relaxation begins from the first snapshot of the DFT substage. For a fair comparison with Uni-Mol+, we also evaluate Uni-Mol+ using both RDKit-generated geometries and geometries after the PM3 and HF optimization. The results of geometry fine-tuning on the random and scaffold splits are shown in Table 4. These results demonstrate that using pre-trained model-relaxed geometries consistently improves downstream property prediction. Furthermore, compared to Uni-Mol+, our approach—combining MLIP-based relaxation with a downstream 3D GNN—achieves superior performance, suggesting that this modular pipeline is more

Table 4: Results of geometry fine-tuning with different kinds of conformers for the downstream PaiNN model on the Molecule3D dataset. In this setting, ground-truth geometries are available during training, while only non-stable geometries are provided at test time.

| Conformer + Model | Random Split | | Scaffold Split | |
|---|---|---|---|---|
| | Validation MAE (eV) | Test MAE (eV) | Validation MAE (eV) | Test MAE (eV) |
| RDKit 3D + Uni-Mol+ | 0.1070 | 0.1090 | 0.1688 | 0.2245 |
| PM3 & HF 3D + Uni-Mol+ | 0.1052 | 0.1080 | 0.1660 | 0.2211 |
| RDKit 3D + PaiNN | 0.1576 | 0.1598 | 0.2089 | 0.2741 |
| PM3 & HF 3D + PaiNN | 0.0889 | 0.0916 | 0.1400 | 0.1880 |
| Force2Geo + PaiNN | **0.0794** | **0.0822** | **0.1281** | **0.1832** |
| DFT 3D + PaiNN (Ground truth) | 0.0562 | 0.0575 | 0.1083 | 0.1548 |

effective than architectures specifically designed to bridge the gap between non-stable and ground truth molecular structures.

### 4.4 Fine-Tuning MLIP Pre-Trained Model for Molecular Property Prediction

**Task.** To demonstrate the effectiveness of the MLIP pre-trained model for 3D molecular property prediction, we fine-tune the pre-trained pre-trained model on downstream tasks where the goal is to predict molecular properties given 3D molecular structures. In this setting, ground-truth geometries are used as input. We evaluate the performance on two benchmark datasets, as described below.

**Datasets.** We use Molecule3D subset created in Section 4.3 and $\nabla^2$DFT (Khrabrov et al., 2024) datasets. The details of the datasets can be found in Appendix E. In this task, we predict the HOMO-LUMO gap, and ground truth 3D geometries are provided. Additional results on the full Molecule3D dataset are provided in Appendix G.1.

Table 5: Results of HOMO-LUMO gap prediction on the $\nabla^2$DFT dataset. Best results are shown in bold, and second-best results are underlined.

| Model | Validation MAE (eV) | Test MAE (eV) |
|---|---|---|
| SchNet | 0.1216 | 0.1461 |
| SphereNet | 0.0625 | 0.0819 |
| ComENet | 0.0831 | 0.1135 |
| DimeNet++ | 0.0545 | 0.0786 |
| TorchMD-Net | 0.0815 | 0.1029 |
| PaiNN | 0.0589 | 0.0857 |
| Force2Prop w/ PaiNN | **0.0483** | **0.0683** |

**Baselines.** We select a set of representative models as baselines, including GIN-virtual (Hu et al., 2021), SchNet (Schütt et al., 2017), SphereNet (Liu et al., 2022), ComENet (Wang et al., 2022), DimeNet++ (Gasteiger et al., 2020), TorchMD-Net (Thölke & De Fabritiis, 2022), Uni-Mol+ (Lu et al., 2023), and PaiNN (Schütt et al., 2021). For the Molecule3D dataset, we include 2D models, 3D models, and hybrid approaches like Uni-Mol+. For the $\nabla^2$DFT dataset, we focus exclusively on 3D models.

**Results.** The results for the Molecule3D random split and scaffold split are reported in Table 6. The fine-tuned pre-trained model achieves the best performance in both settings. The scaffold split is a more challenging evaluation setup, as it requires models to generalize to out-of-distribution molecular scaffolds. As expected, all methods perform worse on the scaffold split compared to the random split, and the gap between validation and test accuracy is also larger in the scaffold setting. GIN-virtual performs the worst among all methods, as it only uses the 2D molecular graph as input, while the HOMO-LUMO gap is highly sensitive to 3D molecular geometry. Uni-Mol+ outperforms GIN-virtual by incorporating ground-truth 3D geometries during training and learning to predict stable 3D structures from RDKit-initialized conformers. However, it still lags behind 3D GNN models, which use

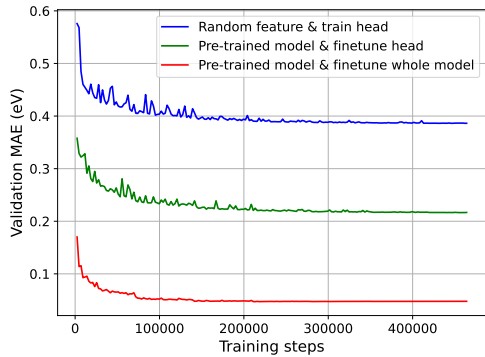

Figure 4: Compare fine-tuning the full pre-trained model versus training only the prediction head using pre-trained or random features.

Table 6: Results of HOMO-LUMO gap prediction on two splits of the Molecule3D dataset. Force2Prop w/ PaiNN denotes the MLIP model pre-trained on our PubChemQCR dataset. Best results are shown in bold, and second-best results are underlined.

| Model | Random Split | | Scaffold Split | |
|---|---|---|---|---|
| | Validation MAE (eV) | Test MAE (eV) | Validation MAE (eV) | Test MAE (eV) |
| GIN-virtual | 0.1249 | 0.1272 | 0.1920 | 0.2421 |
| Uni-Mol+ | 0.1070 | 0.1090 | 0.1688 | 0.2245 |
| SchNet | 0.0718 | 0.0731 | 0.1253 | 0.1837 |
| ComENet | 0.0675 | 0.0693 | 0.1258 | 0.1876 |
| DimeNet++ | 0.0550 | 0.0569 | 0.1106 | 0.1729 |
| TorchMD-Net | 0.0507 | 0.0525 | 0.1037 | 0.1454 |
| PaiNN | 0.0562 | 0.0575 | 0.1083 | 0.1548 |
| Force2Prop w/ PaiNN | **0.0471** | **0.0486** | **0.0911** | **0.1298** |

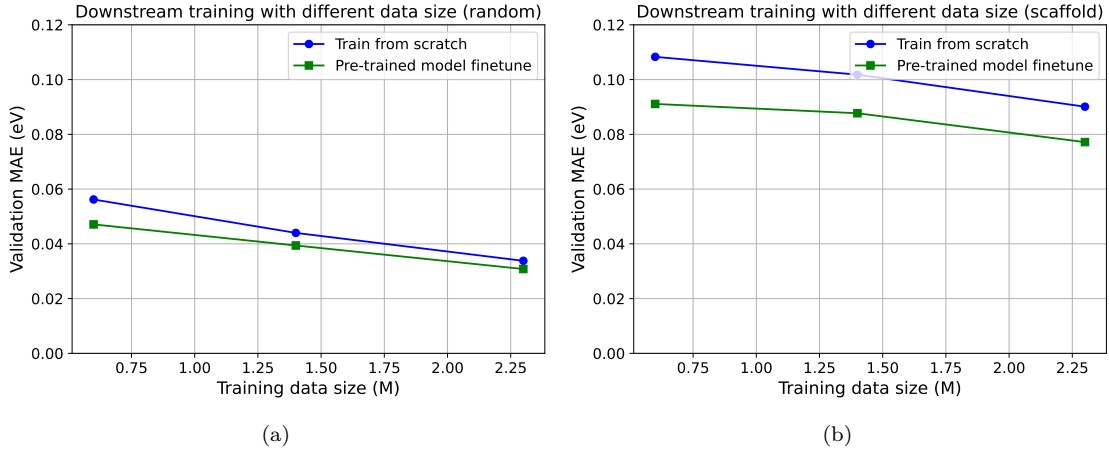

Figure 6: Results of fine-tuning the MLIP pre-trained model on Molecule3D with different sizes of downstream data for (a) random and (b) scaffold split.

accurate 3D geometries during both training and inference. The results on the $\nabla^2$DFT dataset are presented in Table 5. Again, the fine-tuned pre-trained model achieves the best performance among the 3D baselines, further demonstrating its effectiveness in downstream molecular quantum property prediction tasks.

**Analysis.**  Inspired by Zaidi et al. (2022), we conduct an experiment to evaluate the usefulness of pre-trained features. Specifically, we freeze the backbone and fine-tune only the prediction head, and compare it to a baseline where the backbone is randomly initialized and only the prediction head is trained. The training curves are shown in Fig. 4. As expected, fine-tuning only the prediction head performs worse than fine-tuning the entire model, but still significantly outperforms training the head on top of random features. This indicates that pre-trained features provide a meaningful representation for the downstream task. However, to fully adapt the model and achieve optimal performance, it is necessary to fine-tune the entire network.

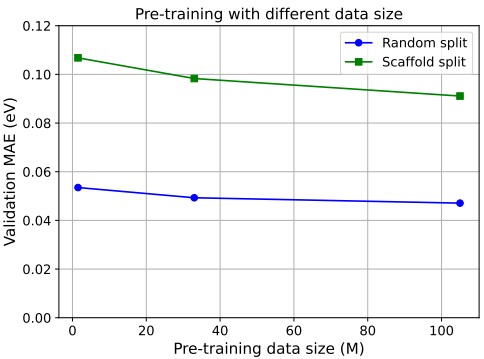

Figure 5: Fine-tuning performance of the pre-trained model pre-trained with different sizes of data.

We also conduct experiments to investigate how the sizes of the pre-training and downstream datasets affect performance on downstream tasks. As a case study, we use HOMO-LUMO gap prediction on the Molecule3D dataset. First, we evaluate the effect of downstream dataset

size by fine-tuning the same pre-trained model on varying amounts of downstream data. The results, shown in Fig. 6, demonstrate that increasing the amount of downstream data consistently improves performance for both the random and scaffold splits. More importantly, the performance gain from fine-tuning the MLIP pre-trained model over training from scratch is especially pronounced when the downstream data is limited. This suggests that the pre-trained models is particularly beneficial in low-data regimes for downstream tasks.

Second, we assess how pre-training dataset size impacts downstream performance by fine-tuning the MLIP pre-trained model pre-trained with different amounts of data while keeping the downstream data fixed. As shown in Fig. 5, downstream accuracy improves steadily as the size of the pre-training dataset increases. These results highlight the dual importance of both pre-training scale and downstream data availability, and underscore the value of pre-trained models in resource-constrained set-

Table 7: Comparison between fine-tuning the pre-trained model and training from scratch using SchNet as the backbone on the Molecule3D dataset.

| Model | Validation MAE (eV) |
|---|---|
| SchNet | 0.0718 |
| Force2Prop w/ SchNet | 0.0572 |

tings. Additionally, we pre-train a different model architecture to demonstrate that the benefits of pre-training generalize across architectures. Specifically, we pre-train SchNet as the pre-trained model and observe that it also outperforms training from scratch, as shown in Table 7, which further validates the effectiveness of the MLIP pre-training.

## 5 Summary

In this work, we train a machine learning interatomic potential (MLIP) pre-trained model using large-scale molecule relaxation data. By curating a dataset comprising 3.5 million small molecules and 300 million snapshots with energy and force labels computed at multiple levels of quantum accuracy, we enable the development of MLIP pre-trained models that can be used to efficiently obtain low-energy 3D structures through geometry optimization for downstream property predictions. Additionally, we introduce geometry fine-tuning as a strategy to mitigate errors and biases introduced during structure relaxation, enhancing the downstream property prediction performance using relaxed 3D geometries. Furthermore, we demonstrate that the MLIP pre-trained model can be directly fine-tuned for molecular property prediction tasks, further extending its applicability.

## 6 Broader Impact Statement

DFT-based geometry optimization can produce highly accurate molecular structures for property prediction, but it is computationally expensive. Our MLIP model, trained on the curated relaxation dataset, demonstrates strong potential for efficiently generating approximate geometries and can improve downstream property prediction to a certain extent. However, we emphasize that the geometries produced by the MLIP model are not yet comparable to those obtained via DFT, and caution should be exercised when applying them in critical or high-stakes scenarios. To support further research, we will release both the curated dataset and the trained model to facilitate continued development of MLIP methods.

## Acknowledgments

SJ acknowledges support from SES AI Corporation, ARPA-H under grant 1AY1AX000053, National Institutes of Health under grant U01AG070112, and National Science Foundation under grant IIS-2243850. XFQ acknowledges support from SES AI Corporation, the Center for Reconfigurable Electronic Materials Inspired by Nonlinear Dynamics (reMIND), an Energy Frontier Research Center funded by the U.S. Department of Energy, Basic Energy Sciences, under Award Number DE-SC0023353. We acknowledge the support of Lambda, Inc. and NVIDIA for providing the computational resources for this project.

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

# A More Details about the Large-Scale DFT Relaxation Dataset

In this section, we introduce more details about our curated PubChemQCR dataset, covering data generation, curation process, and data statistics. Data are publicly available at: `https://huggingface.co/datasets/divelab/PubChemQCR`.

## A.1 Dataset Generation

The raw trajectory data are sourced from the PubChemQC database (Nakata & Shimazaki, 2017). The geometry optimization of each molecule follows a structured protocol. First, initial 3D molecular structures are generated from the InChI representation using OpenBabel, providing the starting point for subsequent quantum calculations. The first optimization is performed using the PM3 semi-empirical method. Next, the relaxed structures undergo further refinement using the Hartree-Fock method with the STO-6G basis set. Finally, the 3D structures are fully optimized using the B3LYP functional with the 6-31G* basis set. PM3 and Hartree-Fock optimizations are performed using the GAMESS software. The DFT optimization consists of three substeps: first, Firefly or SMASH is used for a faster but slightly less accurate optimization. Then, GAMESS is employed for more precise geometry optimization, followed by a final validation step to ensure molecules are indeed optimized.

## A.2 Dataset Curation

The raw trajectory data is 7TB in size and not immediately suitable for machine learning. To make it more accessible, we parsed all raw log files to extract the energies and forces at each snapshot, along with the atomic numbers and coordinates of each atom.

We save the parsed trajectories in six Lightning Memory-Mapped Database (LMDB) files, leveraging its efficient key-value storage and fast data retrieval. Each trajectory is saved as a key-value pair, where the key is the PubChem CID, a unique identifier for chemical compounds in the PubChem database, and the value is a dictionary containing the parsed trajectory, with keys corresponding to the names of each optimization stage, *i.e.* `pm3`, `hf`, `DFT_1st`, and `DFT_2nd`. By storing snapshots from each optimization stage separately, we provide more flexibility in selecting specific trajectory segments for training. The curated dataset is 400GB in size, significantly smaller than the raw data. This size reduction, along with the structured data format, enhances accessibility and usability for the research community, making it easier to develop models using the dataset.

## A.3 Dataset Statistics

The full dataset contains 3,471,000 trajectories and a total of 298,751,667 molecular snapshots, including 163,015,359 snapshots from PM3, 19,274,130 snapshots from Hartree-Fock, 105,494,671 snapshots from the first substage of DFT calculations and 10,967,507 snapshots from the second substage of DFT calculations. On average, each molecule consists of 29 atoms, including 14 heavy atoms, and each molecular trajectory contains approximately 47 PM3 snapshots, 6 Hartree-Fock snapshots, 31 DFT first substage snapshots, and 3 DFT second substage snapshots. This dataset covers 25 chemical elements.

For training efficiency, we also curated a smaller subset for model benchmarking, named PubChemQCR-S, which contains 40,979 trajectories and 1,504,431 molecular snapshots from the first substage of DFT calculations.

# B Benchmarked Methods on PubChemQCR-S

**SchNet** (Schütt et al., 2018) is a continuous-filter convolutional network that models local atomic correlations using learned filter-generating networks. **FAENet** (Duval et al., 2023) introduces frame averaging to enforce symmetry compliance in molecular systems, allowing geometric information to be processed without explicit symmetry-preserving architectural constraints. **NequIP** (Batzner et al., 2022) and **Equiformer** (Liao & Smidt, 2022) are equivariant neural networks that model interatomic interactions while preserving geometric

Table 8: Model configurations—including the number of layers, hidden dimensions (or maximum irreducible representation channels), and batch sizes—are provided for all baseline models trained on the PubChemQCR-S dataset. These models include SchNet (Schütt et al., 2018), PaiNN (Schütt et al., 2021), MACE (Batatia et al., 2022), Equiformer (Liao & Smidt, 2022), PACE (Xu et al., 2024), FAENet (Duval et al., 2023), NequIP (Batzner et al., 2022), SevenNet (Park et al., 2024), and Allegro (Musaelian et al., 2023).

| Models | Layers | Hidden Dimension | Batch Size |
|---|---|---|---|
| SchNet | 4 | 128 | 128 |
| PaiNN | 4 | 128 | 32 |
| FAENet | 4 | 128 | 64 |
| NequIP | 5 | 64 | 16 |
| SevenNet | 5 | 128 | 16 |
| MACE | 2 | 128 | 8 |
| PACE | 2 | 128 | 8 |
| Allegro | 2 | 128 | 8 |
| Equiformer | 4 | 128 | 32 |

symmetries. They maintain various types of geometric features during message passing and capture feature interactions using the Clebsch–Gordan tensor product. Equiformer further incorporates attention mechanisms into the equivariant message passing process. **SevenNet** (Park et al., 2024) is an equivariant model that builds upon the NequIP architecture by introducing a scalable parallelization scheme designed for spatial decomposition in large-scale molecular dynamics (MD) simulations. **Allegro** (Musaelian et al., 2023) is a highly efficient, local equivariant model tailored for scalable, high-accuracy simulations. It models many-body interactions through a series of tensor products of learned equivariant representations. **PaiNN** (Schütt et al., 2021) extends SchNet by introducing equivariant representations, enabling the model to capture directional dependencies while maintaining computational efficiency. **MACE** (Batatia et al., 2022) and **PACE** (Xu et al., 2024) are other equivariant frameworks that capture many-body interactions using symmetry-aware neural architectures.

## C   Training Details of Benchmarked Methods

On the PubChemQCR-S subset, Equiformer uses a separate prediction head to directly predict atomic forces, whereas other methods compute forces as the gradient of the predicted energy. Note that to eliminate the influence of molecular size, we predict the energy per atom rather than the total energy. Additionally, we normalize the energy by subtracting the mean energy during training. To remove the effect of translation, we also center the coordinates by shifting them to have a zero centroid.

Table 8 summarizes the model configurations used for all baseline methods. For FAENet, we adopt the "simple" message-passing variant, while for MACE, we include the residual interaction block to enhance expressiveness. Initial attempts to train Equiformer using its original OC20 settings (6 layers with hidden irreps of either $256{\times}0e + 256{\times}1e$ or $256{\times}0e + 128{\times}1e$) failed to converge; thus, we employ a reduced configuration consisting of 4 layers with irreps $128{\times}0e + 64{\times}1e$. For all tensor-product-based models—including NequIP (Batzner et al., 2022), MACE (Batatia et al., 2022), PACE (Xu et al., 2024), Allegro (Musaelian et al., 2023), SevenNet (Park et al., 2024), and Equiformer (Liao & Smidt, 2022)—only even-parity irreducible representations are used and $L_{max} = 2$ except for Equiformer.

All experiments on the PubChemQCR-S benchmarks employ a cutoff radius of 4.5 Å, the Adam optimizer (Kingma & Ba, 2014) with an initial learning rate of $5 \times 10^{-4}$, and a REDUCELRONPLATEAU learning rate scheduler with a patience of 2 epochs. Models are trained for up to 100 epochs on the PubChemQCR-S subset using NVIDIA A100-80GB GPUs.

## D    Metrics for Geometry Optimization

**Average Energy Minimization Percentage.** This metric quantifies how much energy is minimized by the MLIP-relaxed conformer relative to the DFT-relaxed conformer:

$$\overline{\mathrm{pct}}_T = \frac{1}{|\mathcal{D}_{\mathrm{opt}}|} \sum_{c \in \mathcal{D}_{\mathrm{opt}}} \mathrm{pct}(c_T), \tag{10}$$

where $\mathrm{pct}(c_T)$ is defined as:

$$\mathrm{pct}(c_T) = 100\% \cdot \frac{E_{c_0}^{\mathrm{DFT}} - E_{c_T}^{\mathrm{DFT}}}{E_{c_0}^{\mathrm{DFT}} - E_{c_{\mathrm{gt}}}^{\mathrm{DFT}}}. \tag{11}$$

Here, $c_0$, $c_T$, and $c_{\mathrm{gt}}$ denote the initial conformer, the MLIP-relaxed conformer, and the DFT-relaxed conformer, respectively. $E_{(\cdot)}^{\mathrm{DFT}}$ refers to the single-point DFT energy evaluated at the given conformer.

**Chemical Accuracy Success Rate.** This metric measures the percentage of relaxed conformers whose residual energy is within chemical accuracy (commonly defined as 1 kcal/mol):

$$\mathrm{pct}_{\mathrm{success}} = \frac{1}{|\mathcal{D}_{\mathrm{opt}}|} \sum_{c \in \mathcal{D}_{\mathrm{opt}}} I\left[ E^{\mathrm{res}}(c_T) < 1 \right], \tag{12}$$

with the residual energy defined as:

$$E^{\mathrm{res}}(c_T) = E_{c_T}^{\mathrm{DFT}} - E_{c_{\mathrm{gt}}}^{\mathrm{DFT}}. \tag{13}$$

**Divergence Rate.** This metric, denoted as $\mathrm{pct}_{\mathrm{div}}$, represents the percentage of relaxed molecules for which either the single-point DFT energy calculation failed or the relaxed DFT energy is higher than the initial energy.

**Force Convergence Rate.** This metric measures the percentage of relaxed molecules whose maximum force is below a threshold of 0.05 eV/Å:

$$\mathrm{pct}_{\mathrm{FwT}} = \frac{1}{|\mathcal{D}_{\mathrm{opt}}|} \sum_{c \in \mathcal{D}_{\mathrm{opt}}} I\left[ \max(F(c_T)) < 0.05 \right]. \tag{14}$$

## E    Downstream Datasets Details

**Molecule3D.** The Molecule3D dataset (Xu et al., 2021) is curated from the PubChemQC dataset, which contains nearly 4 million organic small molecules. Each molecule is associated with a ground-state 3D geometry derived from DFT calculations, along with corresponding quantum properties. In our experiments, we focus on predicting the HOMO-LUMO gap, a key quantum property of molecular electronic structure. Molecule3D provides two standard data splits: a *random split*, where training, validation, and test sets are sampled from the same distribution, and a *scaffold split*, which introduces a distribution shift between training and test sets to evaluate model generalization. Since Molecule3D and our curated dataset are derived from the same source, we remove any overlapping molecules from the Molecule3D test set to prevent data leakage.

$\nabla^2$**DFT.** $\nabla^2$DFT (Khrabrov et al., 2024) is a recently introduced large-scale benchmark that includes DFT-level ($\omega$B97X-D/def2-SVP) calculations of energies, forces, molecular properties, and Hamiltonian matrices. For training, we use the large split comprising 99,018 molecules and 500,552 conformations. The structure test split includes 176,001 molecules.

## F    MLIP Pre-Trained Model and Training Details

For the MLIP pre-trained model, we use PaiNN with a hidden dimension of 128, 128 Gaussian components in the radial basis function, 4 layers, and a cutoff distance of 4.5Å. We adopt the PaiNN implementation from the Open Catalyst Project GitHub repository (Chanussot et al., 2021).

Table 9: Results of HOMO-LUMO gap prediction on two splits of the Molecule3D full dataset. Force2Prop w/ PaiNN denotes the MLIP pre-trained model pre-trained on the PubChemQCR dataset. Best results are shown in bold, and second-best results are underlined.

| Model | Random split | | Scaffold split | |
|---|---|---|---|---|
| | Validation MAE (eV) | Test MAE (eV) | Validation MAE (eV) | Test MAE (eV) |
| GIN-virtual | 0.1038 | 0.1053 | 0.1875 | 0.2492 |
| Uni-Mol+ | 0.0849 | 0.0850 | 0.1477 | 0.2044 |
| SchNet | 0.0423 | 0.0438 | 0.0996 | 0.1619 |
| ComENet | 0.0432 | 0.0438 | 0.1048 | 0.1805 |
| DimeNet++ | 0.0398 | 0.0418 | 0.0869 | 0.1451 |
| TorchMD-Net | 0.0348 | 0.0364 | 0.0895 | 0.1403 |
| PaiNN | 0.0338 | 0.0356 | 0.0901 | 0.1378 |
| Force2Prop w/ PaiNN | **0.0308** | **0.0324** | **0.0771** | **0.1204** |

For pre-training the MLIP pre-trained model on PubChemQCR, we use a learning rate of 1e-3 and a batch size of 64. Optimization is performed using Adam (Kingma & Ba, 2014) with $\beta_1 = 0.9$, $\beta_2 = 0.999$, and learning rate scheduling via ReduceLROnPlateau with a patience of 2 epochs. Training is conducted for 9 epochs on four NVIDIA H100 GPUs.

For fine-tuning on downstream tasks, we use a batch size of 256 and a learning rate of 5e-4, with Adam optimizer ($\beta_1 = 0.9$, $\beta_2 = 0.999$). We apply a StepLR scheduler with a step size of 40 and $\gamma = 0.5$, and train on a single NVIDIA A100 GPU.

# G  Additional Results

## G.1  Experimental Results of Molecule3D Full Dataset

Table 9 presents the fine-tuning results on the full Molecule3D dataset. With more training data, all methods show improved performance, and the MLIP pre-trained model continues to achieve the best results, demonstrating the effectiveness of the pre-trained pre-trained model.

## G.2  Different Pre-Training Strategies

In this work, we adopt supervised pre-training to train the pre-trained model for explicit energy and force prediction. Previously, due to the lack of large-scale relaxation datasets with both energy and force labels, prior methods relied on self-supervised pre-training. Among them, denoising pre-training has been the most effective, and we compare our approach against this strategy. Specifically, we consider two denoising methods: coordinate denoising (Zaidi et al., 2022) and the more recent DeNS method (Liao et al., 2024), which is designed to generalize to non-equilibrium structures. We then fine-tune the pre-trained models on the Molecule3D dataset for molecular property prediction. Results in Table 10 show that supervised pre-training outperforms denoising-based methods on downstream tasks, as it enables the model to explicitly learn atomic interactions.

Table 10: Performance of fine-tuning PaiNN pre-trained with different strategies for HOMO-LUMO gap prediction on Molecule3D subset random split.

| Model | Validation MAE (eV) |
|---|---|
| DeNS w/ PaiNN | 0.0560 |
| Denoising w/ PaiNN | 0.0533 |
| Force2Prop w/ PaiNN | **0.0471** |

### G.3 Different Training Hyperparameters in Geometry Fine-Tuning

As described in Section 2.4, we perform geometry fine-tuning using multi-task learning by incorporating an additional geometry alignment task. During training, noise is added to the ground truth conformers to improve generalization. We evaluate the impact of different noise scales, with results shown in Table 11. The results indicate that performance is sensitive to the noise level—both overly small and overly large noise can degrade performance. We also investigate the effect of varying the loss weight for the geometry alignment task. As shown in Table 12, the performance is relatively robust to the choice of loss weight.

Table 11: Performance comparison of different noise scales in multi-task learning for geometry fine-tuning.

| Noise Std | Validation MAE (eV) |
|---|---|
| 0.0 | 0.0876 |
| 0.02 | 0.0817 |
| 0.05 | 0.0807 |
| 0.1 | **0.0794** |
| 0.2 | 0.0825 |
| 0.3 | 0.0867 |
| 0.4 | 0.0907 |

Table 12: Performance comparison of different geometry loss weights in multi-task learning for geometry fine-tuning.

| Geometry Loss Weights | Validation MAE (eV) |
|---|---|
| 0.01 | 0.0799 |
| 0.05 | 0.0802 |
| 0.1 | **0.0794** |
| 0.2 | 0.0801 |
| 0.3 | 0.0808 |

### G.4 Dataset with Only 2D Graphs

In Section 2.4, the task assumes access to ground-truth 3D conformers during training but not at test time. A more challenging setting arises when neither the training nor the test set includes ground truth 3D geometries—only 2D molecular graphs are available. For this experiment, we predict the HOMO-LUMO gap on the QM9 dataset using only 2D graphs. Initial 3D conformers are generated via RDKit and relaxed using our pre-trained model; the downstream predictor is then trained from scratch on these relaxed geometries. As shown in Table 13, a significant performance gap remains between using ground-truth versus relaxed conformers. This is partly because the pre-trained model was trained only on geometries near the energy minimum, making relaxation from RDKit conformers an out-of-distribution task. Also, the geometry optimization capability needs further improvement when it is used to generate stable conformers for downstream property prediction tasks. If the optimized geometries are imperfect, correcting the relaxation bias becomes highly challenging without ground-truth geometries available during training. Although our current pre-trained model does not perform well in this setting, we highlight this scenario as an important future direction, aiming to eliminate reliance on DFT-derived geometries entirely.

Table 13: Performance comparison of downstream predictors trained from scratch on MLIP-relaxed geometries versus ground-truth geometries for the QM9 HOMO-LUMO gap prediction task. In this setting, only 2D molecular graphs are provided for the MLIP to perform relaxation.

| Conformers | Validation MAE (eV) |
|---|---|
| Relaxed 3D | 0.4846 |
| Ground truth 3D | 0.1137 |

