# OpenReview forum: "Augmenting Molecular Graphs with Geometries via Machine Learning Interatomic Potentials"
_TMLR — Accepted by TMLR_

### Review · Reviewer_1nMr · 2025-11-03

**Summary Of Contributions:**

The paper suggests pretraining a MLIP foundation model in order to improve molecular property prediction. First, the authors curated a molecular relaxation dataset and then they trained a MLIP foundation model to predict energy and forces from these structures. Then, the pretrained model can be used in two ways in order to improve property prediction.
1) In case no stable structures are available, the authors relax these structures using the foundation model and then they train the property predictor on the relaxed structures.
2) When ground-truth geometries are available, they directly finetune the foundation model on the downstream task.

Strengths:
1) The authors provide a large-scale relaxation dataset at DFT accuracy that can be proved useful in training molecular foundation models.
2) The authors showcase a consistent improvement of their proposed methods on test datasets and enough ablations to illustrate the improvements introduced by different components.

Weaknesses:
1) The results in Table 3 on geometry optimization of molecules are relatively low suggesting that Force2Geo optimization is far from DFT-level relaxation. While the authors explain the reasons behind the poor results, it would be nice to have some more analysis on this. Perhaps, they can do a similar ablation to Figure 5 but for geometry optimization in order to check the effect of train data size on the accuracy of the relaxation.
2) It is not clearly stated that the dataset will be released in case of publication. I guess this should be the case.
3) I am a bit skeptical about potential data leakage when using Molecule3D dataset. While the authors state that 'Since Molecule3D and our curated dataset are derived from the same source, we remove any overlapping molecules from the Molecule3D test set to prevent data leakage.' very similar molecules may still be present in pretraining and especially when testing on the scaffold split this may revert the distribution shift inserted by scaffold split making the task simpler than it looks.

**Audience:**

Yes

**Audience Explanation:**

Yes.The paper is relevant to the TMLR audience and the authors provide a large-scale relaxation dataset along with a proposed methodology on using a foundation model for property prediction. Many researchers in the area of machine learning would be interested in both the proposed methodology and most importantly the dataset. It is important that this dataset will be released upon publication to promote research on MLIP foundation models. The authors also acknowledge the limitations of the paper in respect of relaxation accuracy and propose ways to improve the performance for future research.

**Broader Impact Concerns:**

Nothing specific.

**Claims And Evidence:**

Yes

**Claims Explanation:**

The answer is yes. The authors showcase that their proposed methods (Force2Geo+PaiNN and Force2Prop+PaiNN) improve the performance of the downstream tasks giving enough ablations. Also, the scaling experiments in Figures 4-6 convincingly back up claims that pretraining helps when downstream data is limited and that larger pretraining datasets improve transferability.

**Requested Changes:**

1) It would be useful to include in Table 2 the training time of each MLIP models in order to justify the selection of PaiNN (instead of Equiformer  that achieves better performance).
2) It would be useful to clarify how the overlap between PubChemQCR and Molecule3D was removed. If it was just removing overlaping structures, the authors should check on how they can remove overlaps but also keep the scaffold split robust with a important distribution shift between the train (+pretraining dataset) and test set.
3) Minor typo on equations (1) and (2). I think $φ_{m}$ and $φ_{u}$ need a superscript $l$ since the mappings are different per layer.
4) I don't understand exactly what is M and what is N in equations 5 and 6. Are these the number of atoms or the number of molecules? The authors should clarify that.

I believe that all of the above recommendations are important for acceptance.

---

> ### Author Response · Authors · 2025-11-20
> **Response -- Part 1**
>
> Thank you for your time and valuable comments! We hope your concerns and questions can be addressed by our responses below.
>
> >**Q1: The results in Table 3 on geometry optimization of molecules are relatively low suggesting that Force2Geo optimization is far from DFT-level relaxation. While the authors explain the reasons behind the poor results, it would be nice to have some more analysis on this. Perhaps, they can do a similar ablation to Figure 5 but for geometry optimization in order to check the effect of train data size on the accuracy of the relaxation.**
>
> - Thanks for the suggestion. We pre-trained PaiNN on a smaller subset and observed that the relaxation performance improves with larger training data. That said, as mentioned in the paper, it remains challenging to train a high-accuracy MLIP using only near-equilibrium data. We believe that incorporating data with broader configurational coverage and more diverse forces would further enhance relaxation performance. However, this is a limitation of the original data source, and we plan to collaborate with the data creators to address this in future work.
>
> | Model | $\overline{pct}_T$ (%)$^\uparrow$ | $pct_{success}$ (%)$^\uparrow$ | $pct_{div}$ (%)$^\downarrow$ | $pct_{FwT}$ (%)$^\uparrow$ | Data Size |
> | ----- | ------ | ------ | ------ | ------ | ------ |
> |`PaiNN`| $57.37$ | $10.29$ | $8.1$ | $4.2$ | $105M$ |
> |`PaiNN`| $25.75$ | $0.1$ | $20.4$ | $0.0$ | $33M$ |
>
>
> >**Q2: It is not clearly stated that the dataset will be released in case of publication. I guess this should be the case.**
>
> - Yes, dataset will be released upon publication.
>
>
> >**Q3: include in Table 2 the training time of each MLIP models in order to justify the selection of PaiNN**
>
> - Thanks for the suggestion. We have included the training time results in the PubChemQCR-S benchmark, as shown below. From the table, we can observe that PaiNN achieves the second-best overall performance. While FAENet is more efficient in training, its performance is not comparable to that of PaiNN. Equiformer attains slightly better accuracy than PaiNN but at a significantly higher computational cost. Considering the trade-off between training efficiency and performance, PaiNN represents a balanced and practical choice for large-scale pre-training. We've updated the manuscript accordingly.
>
> | Model | Time (min/epoch) | Energy MAE (meV/atom) | Force RMSE (meV/Å) |
> | ----- | ------ | ------ | ------ |
> |`SchNet`| $25$ | $5.30$ | $56.55$ |
> |`PaiNN`| $26$ | $5.13$ | $46.34$ |
> |`NequIP`| $130$ | $7.37$ | $54.78$ |
> |`SevenNet`| $150$ | $8.77$ | $47.63$ |
> |`Allegro`| $85$ | $10.86$ | $60.71$ |
> |`FAENet`| $16$ | $7.28$ | $60.24$ |
> |`MACE`| $120$ | $7.54$ | $51.46$ |
> |`PACE`| $140$ | $6.24$ | $50.54$ |
> |`Equiformer`| $65$ | $4.69$ | $34.67$ |

---

> ### Author Response · Authors · 2025-11-20
> **Response -- Part 2**
>
> >**Q4: It would be useful to clarify how the overlap between PubChemQCR and Molecule3D was removed. If it was just removing overlaping structures, the authors should check on how they can remove overlaps but also keep the scaffold split robust with a important distribution shift between the train (+pretraining dataset) and test set.**
>
> - Thanks for the question. Molecule3D contains only the equilibrium (stable) molecular structures, whereas PubChemQCR includes the full relaxation trajectories for each molecule. We first pre-train the model on PubChemQCR using energy and force supervision, and then fine-tune it on Molecule3D for the HOMO–LUMO gap downstream task.
>
> - To avoid data leakage between pre-training and fine-tuning, we cross-check the molecule CIDs between PubChemQCR and the Molecule3D test set, and remove any overlapping molecules from the Molecule3D test split. During fine-tuning, the training split does not contain any molecules from the original test set.
>
> - Even if the removed molecules were added back to the training split, the distribution shift between training and test sets would still remain. Specifically, we compared the distributions among the original train/test split, the train split after adding the overlapping molecules (denote as `Train w/ overlap`), and the test split after removing them (denote as `Test w/o overlap`). We found that both the mean and standard deviation remain similar to the original split, as shown below.
>
> | Split | Mean | Std |
> | ----- | ----- | ----- |
> | `Original Train` | 5.83 | 1.18 |
> | `Original Test` | 5.17 | 1.20 |
> | `Train w/ overlap` | 5.70 | 1.21 |
> | `Test w/o overlap` | 5.10 | 1.21 |
>
> - Additionally, as shown in Table 6, there is a substantial performance gap between the random and scaffold splits for all methods, further indicating that a distribution shift still exists between the two splits. Moreover, during pre-training, the model is trained only on energy and force, not on the HOMO–LUMO gap. In summary, there is no data leakage, and the key distribution differences remain valid.
>
> >**Q5: Minor typo on equations (1) and (2).**
>
> - Thanks for pointing out. We should add superscript l to represent layer. We've updated the manuscript accordingly/
>
> >**Q6: I don't understand exactly what is M and what is N in equations 5 and 6. Are these the number of atoms or the number of molecules? The authors should clarify that.**
>
> - N denotes the number of molecules in a batch, M denotes the number of atoms in a batch. We've updated the manuscript accordingly.

---

### Review · Reviewer_Pk1o · 2025-12-18

**Summary Of Contributions:**

This paper addresses the challenge of obtaining accurate three-dimensional molecular geometries for molecular property prediction when density functional theory based geometries are unavailable or prohibitively expensive. The main contribution is the curation and use of PubChemQCR, a large-scale molecular relaxation dataset derived from PubChemQC, containing approximately 3.5 million molecules and nearly 300 million molecular snapshots with energy and force labels, including over 100 million snapshots computed at the DFT B3LYP/6-31G* level.

Using this dataset, the authors train a supervised machine learning interatomic potential foundation model, primarily based on a PaiNN backbone, to predict molecular energies and forces. The pretrained model is then used in three distinct but related ways. First, it is employed as a surrogate potential for geometry optimization, producing relaxed molecular conformers that can be fed into downstream three-dimensional graph neural networks for property prediction (Force2Geo). Second, the authors introduce a geometry fine-tuning strategy that adapts downstream predictors to the distribution shift induced by MLIP-relaxed geometries via a multi-task objective that includes a geometry alignment loss. Third, the pretrained MLIP model itself is fine-tuned directly for molecular property prediction when ground-truth geometries are available (Force2Prop).

The key strengths of the work are the scale and engineering effort involved in dataset construction, the modular framing that separates geometry generation from downstream prediction, and the comprehensive experimental evaluation across geometry optimization and property prediction benchmarks. The strongest empirical results are in the Force2Prop setting, where consistent improvements over strong baselines are reported. The main weakness is that the geometry optimization performance remains weak in absolute terms, and the current results do not convincingly demonstrate that MLIP-based relaxation can serve as a reliable replacement for DFT when no ground-truth geometries are available.

**Additional Comments:**

I personally believe this is an ambitious and technically substantial paper that makes a meaningful contribution through dataset curation and careful experimental evaluation. The Force2Prop results are strong and convincingly demonstrate the value of supervised MLIP pretraining for molecular property prediction. The Force2Geo results are more exploratory and currently fall short of supporting the broader motivation of replacing DFT-based geometry optimization.

If the paper is reframed to emphasize representation learning and property prediction more strongly, while presenting geometry optimization as an important but unresolved challenge, it would be a solid and informative contribution to TMLR.

I want to note that my background in quantum chemistry and interatomic potentials is limited, and I had to rely on careful reading of the related work to form this assessment. It is possible that I have missed or misunderstood some aspects of the work, and I remain open to engaging with the authors to clarify these points.

**Audience:**

Yes

**Audience Explanation:**

This work addresses a problem of central importance to machine learning for chemistry, drug discovery, and geometric deep learning. The lack of large-scale, force-labeled datasets for small molecules is a well-known bottleneck, and the PubChemQCR dataset alone is likely to be of significant interest to the TMLR audience.

Beyond the dataset, the paper contributes to ongoing discussions about molecular foundation models and the relative merits of supervised, physics-grounded pretraining versus self-supervised denoising approaches. The modular framing of Force2Geo and Force2Prop provides a useful conceptual lens for thinking about how MLIP models can be integrated into downstream pipelines. Even the negative and mixed results for geometry optimization are informative, as they highlight fundamental challenges in learning near-equilibrium force fields and generalizing beyond the training distribution.

I believe people interested in AI for science, molecular representation learning, and scalable pretraining strategies will likely find both the successes and limitations of this work valuable.

**Broader Impact Concerns:**

The work does not raise major ethical concerns beyond those typical for computational chemistry research. However, a brief Broader Impact Statement would strengthen the paper by addressing the computational cost of large-scale DFT computation and model training, as well as the risks of over-reliance on approximate geometries in safety-critical applications such as drug discovery. Discussion of the dataset and model release plans would also support transparency and reproducibility.

**Claims And Evidence:**

No

**Claims Explanation:**

The claim that large-scale supervised pretraining on relaxation trajectories yields transferable MLIP representations is well supported. The Force2Prop results on Molecule3D and ∇2DFT consistently outperform strong three-dimensional baselines such as SchNet, DimeNet++, TorchMD-Net, and PaiNN. The ablation studies on pretraining data size, downstream data size, and fine-tuning method convincingly show that pretraining is beneficial, especially in low-data regimes. Comparisons against denoising-based pretraining further support the argument that explicit energy and force supervision leads to stronger representations.

However, the claim that the foundation model can provide valuable molecular geometries via geometry optimization is only partially supported. The geometry optimization metrics in Table 3 show that only 10.29 percent of molecules reach chemical accuracy and only 4.2 percent satisfy the force convergence criterion. The gap between Force2Geo-relaxed geometries and DFT-relaxed geometries remains large, with downstream property prediction errors roughly 40 percent higher than when using DFT geometries. While the authors are transparent about these limitations and provide a qualitative discussion of why near-equilibrium force prediction is difficult, the evidence does not yet justify strong claims about the practical reliability of MLIP-based relaxation in this setting.

Overall, the experimental evidence is accurate and extensive, but it fully supports the Force2Prop and representation learning claims more convincingly than the geometry optimization claims that motivate the Force2Geo pipeline.

**Note** - My assessment of the claims is mixed rather than strictly negative. Several claims, especially those related to representation learning and the Force2Prop setting, are well supported by the experiments. However, claims about geometry optimization are less convincingly supported by the current results. Because the form allows only a Yes or No response, I selected No to reflect this imbalance, and I am open to discussion with the authors.

**Requested Changes:**

## Critical Changes

### 1. More Careful Framing of Geometry Optimization Claims
The abstract and introduction should more clearly reflect the limitations of the Force2Geo results. Statements suggesting that the model provides generally reliable geometries should be softened to align with the reported chemical accuracy and convergence rates.

### 2. Computational Cost Analysis
The paper claims that MLIP-based relaxation is a cost-effective alternative to DFT, but no quantitative comparison is provided. A concrete analysis of wall-clock time and computational cost for MLIP-based relaxation versus standard DFT relaxation is required to justify this claim.

### 3. Baseline Comparisons for Geometry Optimization
The geometry optimization results should be contextualized by comparison with simpler baselines such as classical force fields (e.g., UFF or MMFF). Without this, it is difficult to assess whether MLIP-based relaxation offers a meaningful advantage.

### 4. Dataset Splitting and Leakage Details
The paper should more clearly describe how the foundation model training, validation, and test splits were constructed, and how overlap with downstream benchmarks was detected and removed.

---

## Changes That Would Strengthen the Work

### 1. Failure Analysis for Geometry Optimization
More detailed empirical analysis of failed optimization cases would be valuable. This could include visualizations of relaxation trajectories, breakdowns by molecular size or functional group, or analysis of force prediction errors across different force regimes.

### 2. Use of Multi-Fidelity Data
Since PubChemQCR includes PM3, HF, and DFT stages, exploring multi-fidelity or curriculum-based training strategies could potentially improve Force2Geo performance and would strengthen the methodological contribution.

### 3. Justification of Backbone Choice
Equiformer achieves stronger force prediction accuracy than PaiNN in Table 2. A brief discussion or experiment quantifying performance versus computational cost would better justify the architectural choice.

### 4. Visualization of Geometry Fine-Tuning Effects
Qualitative examples comparing RDKit, MLIP-relaxed, and DFT geometries for both success and failure cases would provide intuition for why geometry fine-tuning improves downstream performance.

---

> ### Author Response · Authors · 2025-12-28
> **Response -- Part 1**
>
> Thank you for your time and valuable comments! We hope your concerns and questions can be addressed by our responses below.
>
> >**Q1: However, the claim that the foundation model can provide valuable molecular geometries via geometry optimization is only partially supported. The geometry optimization metrics in Table 3 show that only 10.29 percent of molecules reach chemical accuracy and only 4.2 percent satisfy the force convergence criterion. The gap between Force2Geo-relaxed geometries and DFT-relaxed geometries remains large, with downstream property prediction errors roughly 40 percent higher than when using DFT geometries.**
>
> - We acknowledge that the geometry optimization performance is not yet ideal. This limitation is largely due to constraints in the current dataset, as discussed in the manuscript. Nevertheless, the model is already effective at generating relaxed structures that improve downstream property prediction, which is the primary goal of this work. As shown in Table 4, using MLIP-relaxed geometries as input to the downstream predictor significantly narrows the performance gap relative to using ground-truth geometries. We agree that further improvements in geometry optimization would likely lead to additional gains in downstream accuracy. Such improvements could be achieved by adopting more expressive backbone models and expanding the dataset to include complete DFT-based relaxation trajectories with broader coverage of the potential energy surface. We plan to pursue these directions in future work.
>
> >**Q2: *More Careful Framing of Geometry Optimization Claims*: The abstract and introduction should more clearly reflect the limitations of the Force2Geo results. Statements suggesting that the model provides generally reliable geometries should be softened to align with the reported chemical accuracy and convergence rates.**
>
> - Thank you for the suggestion. We have revised the abstract and introduction to more explicitly acknowledge that the geometry optimization results are approximate and do not consistently reach DFT-level chemical accuracy, while still being useful for downstream property prediction to some extent.
>
> >**Q3: *Computational Cost Analysis*: The paper claims that MLIP-based relaxation is a cost-effective alternative to DFT, but no quantitative comparison is provided. A concrete analysis of wall-clock time and computational cost for MLIP-based relaxation versus standard DFT relaxation is required to justify this claim.**
>
> - Below we report the average wall-clock time for MLIP-based and DFT-based geometry relaxation on our dataset. MLIP-based relaxation is orders of magnitude more efficient than DFT-based relaxation, although its accuracy still requires improvement. These results demonstrate the potential of MLIP-based methods as a cost-effective alternative to DFT for large-scale applications.
>
> | Method | Average relaxation time |
> | ----- | ----- |
> | `DFT` | 1.72 hrs |
> | `MLIP` | 0.48 s |
>
> >**Q4: *Baseline Comparisons for Geometry Optimization*: The geometry optimization results should be contextualized by comparison with simpler baselines such as classical force fields (e.g., UFF or MMFF). Without this, it is difficult to assess whether MLIP-based relaxation offers a meaningful advantage.**
>
> - In Table 4, the RDKit 3D baseline already applies UFF-based relaxation. The improved downstream performance achieved using MLIP-relaxed geometries therefore indicates that MLIP-based relaxation provides more informative geometries than classical force-field baselines for the evaluated property prediction task.
>
> >**Q5: *Dataset Splitting and Leakage Details*: The paper should more clearly describe how the foundation model training, validation, and test splits were constructed, and how overlap with downstream benchmarks was detected and removed.**
>
> - For the PubChemQCR-S subset, we split the data into training, validation, and test sets using a 60%/20%/20% split. For the full PubChemQCR dataset, we retain the same test set as PubChemQCR-S and divide the remaining data into training and validation sets using an 80%/20% split. To prevent data leakage between pre-training and downstream evaluation, we cross-check molecular CIDs between PubChemQCR and the Molecule3D test set and remove any overlapping molecules from the Molecule3D test split. During fine-tuning, the training data does not contain molecules from the downstream test set.

---

> ### Author Response · Authors · 2025-12-28
> **Response -- Part 2**
>
> >**Q6: *Justification of Backbone Choice*: Equiformer achieves stronger force prediction accuracy than PaiNN in Table 2. A brief discussion or experiment quantifying performance versus computational cost would better justify the architectural choice.**
>
> - Thank you for the suggestion. We have added training-time comparisons on the PubChemQCR-S benchmark. PaiNN achieves the second-best overall accuracy while maintaining significantly lower computational cost than Equiformer. Although Equiformer attains stronger force prediction accuracy, its training cost is substantially higher. Considering the trade-off between efficiency and performance, PaiNN provides a balanced and practical backbone for large-scale pre-training. We will update the manuscript accordingly.
>
> | Model | Time (min/epoch) | Energy MAE (meV/atom) | Force RMSE (meV/Å) |
> | ----- | ------ | ------ | ------ |
> |`SchNet`| $25$ | $5.30$ | $56.55$ |
> |`PaiNN`| $26$ | $5.13$ | $46.34$ |
> |`NequIP`| $130$ | $7.37$ | $54.78$ |
> |`SevenNet`| $150$ | $8.77$ | $47.63$ |
> |`Allegro`| $85$ | $10.86$ | $60.71$ |
> |`FAENet`| $16$ | $7.28$ | $60.24$ |
> |`MACE`| $120$ | $7.54$ | $51.46$ |
> |`PACE`| $140$ | $6.24$ | $50.54$ |
> |`Equiformer`| $65$ | $4.69$ | $34.67$ |
>
>
> >**Q7 *Use of Multi-Fidelity Data*: Since PubChemQCR includes PM3, HF, and DFT stages, exploring multi-fidelity or curriculum-based training strategies could potentially improve Force2Geo performance and would strengthen the methodological contribution.**
>
> - Actually, the energy and force labels obtained at different stages of the same trajectory exhibit varying levels of accuracy and are neither directly comparable nor mutually consistent. Consequently, when training MLIP, it is better to utilize only the DFT-optimized segments, as they provide the highest fidelity labels. For the future work, to ensure label consistency throughout the optimization trajectory, we'd like to collaborate with data providers to generate the entire geometry optimization using a single, uniform level of DFT functional and basis set.
>
> >**Q8 *Failure Analysis for Geometry Optimization and Visualization of Geometry Fine-Tuning Effects*: More detailed empirical analysis of failed optimization cases would be valuable. This could include visualizations of relaxation trajectories, breakdowns by molecular size or functional group, or analysis of force prediction errors across different force regimes. Qualitative examples comparing RDKit, MLIP-relaxed, and DFT geometries for both success and failure cases would provide intuition for why geometry fine-tuning improves downstream performance.**
>
> - We thank the reviewer for these valuable suggestions. While a detailed failure analysis would provide deeper insights into model behavior, it is beyond the scope of the current work. We plan to include such analyses in future studies. That said, the current experiments already demonstrate that pre-training MLIP models on large-scale relaxation data benefits downstream property prediction, both via geometry fine-tuning and direct fine-tuning.
>
> >**Q9: The work does not raise major ethical concerns beyond those typical for computational chemistry research. However, a brief Broader Impact Statement would strengthen the paper by addressing the computational cost of large-scale DFT computation and model training, as well as the risks of over-reliance on approximate geometries in safety-critical applications such as drug discovery. Discussion of the dataset and model release plans would also support transparency and reproducibility.**
>
> - Thank you for the suggestion. We have added a Broader Impact Statement to the manuscript. We emphasize that while MLIP-based geometry optimization offers substantial computational savings, the generated geometries remain approximate and should be used with caution in safety-critical applications. We also confirm our plan to release both the curated dataset and trained models to support transparency and reproducibility.
>
>
> >**Q10: If the paper is reframed to emphasize representation learning and property prediction more strongly, while presenting geometry optimization as an important but unresolved challenge, it would be a solid and informative contribution to TMLR.**
>
> - Thank you for the suggestion and support. We have revised the abstract and introduction to place greater emphasis on representation learning and property prediction, while presenting geometry optimization as an important but still unresolved challenge.

---

### Review · Reviewer_kKwo · 2025-12-22

**Summary Of Contributions:**

The paper demonstrates an ambitious workflow combining ML-based geometry optimization of molecules with ML-based molecular property prediction. MLIP models are trained on a large molecular relaxation dataset and are then applied to geometry optimization and fine-tuned for molecular property prediction tasks. The geometry fine-tuning step introduced in the paper is an interesting contribution to improve the overall performance of the proposed workflow. If the dataset could be made available, it would also be a major contribution and a valuable resource for the community.

Applying graph neural network models (called 3DGNNs in the paper) trained to predict total energy and atomic forces for geometry optimization (or molecular dynamics) or to predict molecular properties directly is not new, but it is interesting to see these tasks combined in a workflow. However, the results presented in the paper are not entirely convincing and important information required to assess the results is either missing or needs clarification before the paper is ready for publication (see detailed comments below).

**Audience:**

Yes

**Audience Explanation:**

Many related works describe how ML-based techniques can accelerate drug or material discovery by replacing computationally expensive methods, but they often focus on a single task without considering the effect on downstream tasks. The application of MLIP-based geometry optimization followed by ML-based property prediction proposed in this paper is ambitious and interesting.

**Broader Impact Concerns:**

None.

**Claims And Evidence:**

No

**Claims Explanation:**

There is not enough information about the training and evaluation procedures of all the different models to fully assess or reproduce the results presented in the paper (see detailed comments below).

**Requested Changes:**

In the introduction, please clarify what is seen as the main contributions of the paper, i.e. what is novel and what is not. This will help the reader navigate the paper and strengthen the work.

The MLIP model presented in this paper is described as a “foundation model”. Without showing generalization to a wide range of tasks, I don’t think there is sufficient support for this claim. Using the term “foundation model” should be justified or changed.

As a reader, I am wondering what model architectures are applied in the work, but these are not introduced before section 4. This information could be mentioned in the introduction, directly or by adding a reference to a subsequent section.

In section 2.2, the notation E is used both as the total energy and as the adjacency matrix. Please consider using different notation.

The training objective for the MLIP model, defined in section 2.2, combines absolute error on energy and squared error on forces. What is the reasoning for combining absolute and squared error? Is this objective used for all the MLIP models evaluated in the paper? If this decision is based on any previous work, please add a reference.

Section 3 on related work briefly covers models based on 1D SMILES strings and 2D molecular graphs but does not mention important work about models based on 3D structures, which are arguably more directly related to the present work. Please mention some of the more directly related work in this section or point to a section where this information can be found.

In section 4, it is difficult to follow exactly what models were used when and how they were trained. For example, the capacity of a NequIP or MACE model varies significantly with the node size and the L-order. Please clarify and make sure training details are presented in the paper or in a referenced appendix. Specifically, it is not completely clear from the main text what exact model architectures are referred to as “Force2Geo” and “Force2Prop” and how they were trained.

In section 4.2 it is noted that the performance of the geometry optimization is not ideal and the training data primarily has low forces. How does the distribution of forces in the training data compare to the distribution of forces in the test data, specifically the initial structures for the relaxation? Perhaps the training data is not sufficient for the task and more data with higher forces (for example from molecular dynamics simulations) is needed?

In section 4.3 it is not entirely clear if and how the geometry fine-tuning approach introduced in section 2.4 is applied and how it affects performance. It would be interesting to see results of a property prediction from MLIP relaxed geometries using a property prediction model trained with and without the geometry fine-tuning step to assess the effect of geometry fine-tuning.

The suite of models that are evaluated in section 4.1 and section 4.4 are not the same. Please clarify how and why were these models selected for each task?

In figure 4, how do the fine-tuning results compare to training a model from scratch?

In figures 5 and 6, please consider fixing the lower limit on the y-axis to zero to make the presentation of the results clearer. Showing such a narrow range on the y-axis can make the relative differences look more dramatic than they really are and thus be misleading. Using the same limits in all three plots would also make the results easier to compare and interpret.

---

> ### Author Response · Authors · 2025-12-28
> **Response -- Part 1**
>
> Thank you for your time and valuable comments! We hope your concerns and questions can be addressed by our responses below.
>
> >**Q1: In the introduction, please clarify what is seen as the main contributions of the paper, i.e. what is novel and what is not. This will help the reader navigate the paper and strengthen the work.**
>
> - We have updated the introduction to clearly outline the main contributions:
>
>   - We curate a large-scale molecular relaxation dataset with 3.5M molecules and 300M snapshots, including 105M DFT-level energy and force labels, enabling MLIP model pre-training.
>   - We show that the pre-trained MLIP model on our dataset can efficiently produce low-energy 3D geometries via geometry optimization for downstream property prediction, and introduce geometry fine-tuning to further improve 3D GNN performance.
>   - We demonstrate that the pre-trained model can be directly fine-tuned for molecular property prediction when ground-truth 3D geometries are available, highlighting the effectiveness of pre-trained MLIPs in supporting diverse downstream tasks.
>
> >**Q2: The MLIP model presented in this paper is described as a “foundation model”. Without showing generalization to a wide range of tasks, I don’t think there is sufficient support for this claim. Using the term “foundation model” should be justified or changed.**
>
> - We thank the reviewer for this comment and have replaced the term “foundation model” with “pre-trained MLIP model” throughout the paper to better reflect the scope of our work. While the model supports multiple downstream uses, we agree that “pre-trained” more accurately characterizes its capabilities without overstatement.
>
>
> >**Q3: As a reader, I am wondering what model architectures are applied in the work, but these are not introduced before section 4. This information could be mentioned in the introduction, directly or by adding a reference to a subsequent section.**
>
> - We have added references in the introduction pointing to Section 4, where model architectures are described in detail.
>
> >**Q4: In section 2.2, the notation E is used both as the total energy and as the adjacency matrix. Please consider using different notation.**
>
> - We have corrected the notation by replacing the adjacency matrix symbol E with A.
>
>
> >**Q5: The training objective for the MLIP model, defined in section 2.2, combines absolute error on energy and squared error on forces. What is the reasoning for combining absolute and squared error? Is this objective used for all the MLIP models evaluated in the paper? If this decision is based on any previous work, please add a reference.**
>
> - Sorry for causing the confusion. This is a typo and it should be RMSE loss for force. We've updated the manuscript to correct that.
>
> >**Q6: Section 3 on related work briefly covers models based on 1D SMILES strings and 2D molecular graphs but does not mention important work about models based on 3D structures, which are arguably more directly related to the present work. Please mention some of the more directly related work in this section or point to a section where this information can be found.**
>
> - We have expanded Section 3 to include additional 3D GNN models commonly used for molecular property prediction with ground-truth geometries, as suggested. Previously in section 3, we mainly mentioned the pre-trained model in this field. There're not so many 3D pre-trained model previously as there's no such large-scale labeled 3D dataset before. Most pre-training models are 3D denoising models and we've already mentioned in the manuscript.
>
> >**Q7: In section 4, it is difficult to follow exactly what models were used when and how they were trained. For example, the capacity of a NequIP or MACE model varies significantly with the node size and the L-order. Please clarify and make sure training details are presented in the paper or in a referenced appendix. Specifically, it is not completely clear from the main text what exact model architectures are referred to as “Force2Geo” and “Force2Prop” and how they were trained.**
>
> - Thanks for the suggestion. We have added detailed architecture and training configurations in Appendix C, including irreps, L-order, learning rates, epochs, and optimizers, etc. Force2Geo and Force2Prop refers how we use the pre-trained MLIP model for downstream property prediction. The backbone model is PaiNN, as described in the section 4.1. Force2Geo means we use pre-trained MLIP model to perform relaxation to provide geometries for downstream predictor to predict properties. Force2Prop means the pre-trained MLIP model is directly fine-tuned for downstream property prediction where the 3D geometry is already given.

---

> ### Author Response · Authors · 2025-12-28
> **Response -- Part 2**
>
> >**Q8: In section 4.2 it is noted that the performance of the geometry optimization is not ideal and the training data primarily has low forces. How does the distribution of forces in the training data compare to the distribution of forces in the test data, specifically the initial structures for the relaxation? Perhaps the training data is not sufficient for the task and more data with higher forces (for example from molecular dynamics simulations) is needed?**
>
> - For the pre-training dataset, we random split the train, validation, and test set, so the force distribution between them are similar. The low-force constraint is caused by the limitation of current dataset. For future work, we will work with our collaborator to perform the entire geometry optimization using a single, uniform level of DFT functional and basis set. Thus, we can have more diverse force magnitudes and broader sampling of molecular conformational space.
>
> >**Q9: In section 4.3 it is not entirely clear if and how the geometry fine-tuning approach introduced in section 2.4 is applied and how it affects performance. It would be interesting to see results of a property prediction from MLIP relaxed geometries using a property prediction model trained with and without the geometry fine-tuning step to assess the effect of geometry fine-tuning.**
>
> - Below we compare the performance with and without geometry fine-tuning. The results show that geometry fine-tuning consistently improves performance, validating its effectiveness in enabling the downstream predictor to adapt to the geometric distribution produced by the pre-trained MLIP model.
>
> | Model | Validation MAE (eV) | Test MAE (eV) |
> | ----- | ------ | ------ |
> |`Without geometry fine-tuning`| $0.0101$ | $0.1052$ |
> |`With geometry fine-tuning`| $0.0794$ | $0.0822$ |
>
> >**Q10: The suite of models that are evaluated in section 4.1 and section 4.4 are not the same. Please clarify how and why were these models selected for each task?**
>
> - Section 4.1 compares different MLIP backbone choices for pre-training, with the goal of identifying a model that achieves strong energy and force prediction accuracy while remaining computationally efficient to train. The models evaluated in Section 4.1 are widely used MLIP architectures in the community. In contrast, Section 4.4 focuses on downstream molecular property prediction, where the evaluated models are commonly used 2D and 3D GNNs for the corresponding tasks, as adopted in prior work.
>
> >**Q11: In figure 4, how do the fine-tuning results compare to training a model from scratch?**
>
> - This comparison is already included in Table 6 and is reproduced below for clarity. The results show that fine-tuning consistently outperforms training from scratch, demonstrating the effectiveness of the pre-trained model.
>
> | Model | Validation MAE (eV) | Test MAE (eV) |
> | ----- | ------ | ------ |
> |`Train from scratch`| $0.0562$ | $0.0575$ |
> |`Fine-tuning`| $0.0471$ | $0.0486$ |
>
>
> >**Q12: In figures 5 and 6, please consider fixing the lower limit on the y-axis to zero to make the presentation of the results clearer. Showing such a narrow range on the y-axis can make the relative differences look more dramatic than they really are and thus be misleading. Using the same limits in all three plots would also make the results easier to compare and interpret.**
>
> - Thank you for the suggestion. We initially zoomed in on the plots to improve readability by allowing them to fit the figure. Following the reviewer’s recommendation, we have now fixed the y-axis limits of all three plots to
> [0, 0.12], which we believe makes the comparisons clearer and easier to interpret.

---

> ### Comment · Reviewer_kKwo · 2026-01-07
>
> Thank you for the thorough response to all comments. The revised manuscript is a big improvement. I only have two minor comments to follow up:
>
> * Appendices B and C are not referenced from the main text, as far as I can tell. Adding references in section 4.1 would help the reader find this information.
> * The updated figures 4 and 6 still say “foundation model” in the legend.

---

> > ### Author Response · Authors · 2026-01-08
> >
> > We’re glad to hear that the comments have been resolved. We have updated the manuscript to add references to Appendices B and C in Section 4.1, and we have also updated the legends of Figures 4 and 6 to use the term “pre-trained model.”

---

### Decision · Action_Editor_RGtv · 2026-02-05

**Recommendation:** Accept as is

**Audience:**

Yes

**Audience Explanation:**

Surrogate energy calculations for molecules obviously makes a lot of sense. It is a very active research topic in machine learning.

**Claims And Evidence:**

Yes

**Claims Explanation:**

Yes, the authors have clearly demonstrated experimentally that the methods they propose work.